# STING Targeting in Lung Diseases

**DOI:** 10.3390/cells11213483

**Published:** 2022-11-03

**Authors:** Dorian de Moura Rodrigues, Norinne Lacerda-Queiroz, Isabelle Couillin, Nicolas Riteau

**Affiliations:** 1Experimental and Molecular Immunology and Neurogenetics Laboratory, University of Orleans, Centre National de la Recherche Scientifique (CNRS), UMR7355, 45100 Orleans, France; 2Key-Obs: Preclinical CRO, 45100 Orleans, France

**Keywords:** lung, STING, adjuvant, vaccine

## Abstract

The cGAS–STING pathway displays important functions in the regulation of innate and adaptive immunity following the detection of microbial and host-derived DNA. Here, we briefly summarize biological functions of STING and review recent literature highlighting its important contribution in the context of respiratory diseases. Over the last years, tremendous progress has been made in our understanding of STING activation, which has favored the development of STING agonists or antagonists with potential therapeutic benefits. Antagonists might alleviate STING-associated chronic inflammation and autoimmunity. Furthermore, pharmacological activation of STING displays strong antiviral properties, as recently shown in the context of SARS-CoV-2 infection. STING agonists also elicit potent stimulatory activities when used as an adjuvant promoting antitumor responses and vaccines efficacy.

## 1. Introduction 

Stimulator of interferon genes (STING) also known as MITA/MPYS/ERIS encoded by *Tmem173* gene is an evolutionary conserved transmembrane protein found in most animal species, including chordates, insects, mollusks and cnidarians, and whose primary role is to sense cytoplasmic DNA [1]. STING was first described by Hiroki Ishikawa and Glen Barber in 2008 as a pattern recognition receptor (PRR) family member of the innate immune system [2]. Inactivated STING resides within the endoplasmic reticulum (ER) membrane. It contains four transmembrane (TM) regions followed by a cytoplasmic ligand-binding domain (LBD) and a C-terminal tail (CTT) [3]. Inactive STING forms a dimer, auto-inhibited by an intramolecular interaction between the LBD and the CTT. Ligand binding induces a conformational change leading to the formation of STING tetramers and oligomers [3] and the CTT domain of STING serves as docking sites for TBK-1 and IRF-3 which undergo phosphorylation [4,5]. STING requires palmitoylation, i.e., covalent addition of palmitic acid at cysteine residues (Cys88 and Cys91), for its multimerization and function. [6]. Active STING translocates from the ER to the ER-Golgi intermediate compartment (ERGIC) and the Golgi [7]. Reports have shown that IRF3 activation occurs in the ERGIC [8] and that sulfated glycosaminoglycans (sGAGs) bind to luminal STING residues in the Golgi to promote STING and TBK1 polymerization and activation [9]. STING also activates nuclear factor κB (NF-κB) [10]. IRF3 and NF-kB act as transcription factors to induce the production of type I interferons (IFNs) and other cytokines involved in host immune responses. 

STING is typically activated by cyclic dinucleotides (CDN). A CDN consists of two nucleotides (e.g., adenine or guanine) connected by phosphodiester bonds in two positions to form a cyclic molecule [11]. CDN activating STING can be exogenous or endogenous. Exogenous CDN are second messengers central to various bacterial processes including metabolism and virulence [11] and, upon infection, these CDN, such as bis-(3′,5′)-cyclic diguanosine monophosphate (c-di-GMP), bind and activate STING [12]. The endogenous CDN activating STING is 2′3′-cyclic GMP-AMP (cGAMP), which is characterized by a noncanonical structure due to the presence of the atypical 2′-5′ phosphodiester linkage between the guanosine and the adenosine. cGAMP is synthesized by the enzyme cyclic GMP-AMP synthase (cGAS) in response to binding either host- or pathogen-derived cytosolic double stranded (ds) DNA [13,14,15]. cGAS is an evolutionary conserved protein with, for instance, cGAS-like receptors identified in Drosophila, activating STING- and NF-κB-dependent antiviral immunity in response to infection with RNA or DNA viruses [16,17]. Recent data suggest that the cGAS/STING pathway originated in bacteria and that cGAMP signaling confers protection against phage infection [18,19]. 

STING pathway is commonly triggered by a wide range of pathogens. STING is critical for host defense against various DNA and RNA viruses. STING-deficient mice are susceptible to lethal infection by herpes simplex virus (HSV-1) and vesicular stomatitis virus (VSV) [20]. In contrast, while STING has been shown to elicit mild bacterial burden control after acute infection by *Listeria monocytogenes* [21], it was latter shown that STING-dependent type I IFNs are detrimental due to their impedance of cell-mediated immunity and that STING-deficient mice are protected against secondary challenge [22]. Further, DNA from the extracellular bacterial pathogen *Streptococcus pneumoniae* stimulates STING-dependent type I IFNs that favor bacterial clearance [23]. It remains unclear why STING exerts opposite functions upon bacterial infection, potentially reflecting specific contributions of type I IFN on bacterial survival, invasive capacity or anti-bacterial immunity. It has also been shown that genomic DNA from protozoan parasites, including plasmodium species, triggers the cGAS–STING pathway [24]. In summary, STING is activated by a wide range of pathogens with contrasting outcomes for the host. The importance of STING signaling in host defense is highlighted by the fact that numerous pathogens have developed strategies to interfere with its function [25]. 

Besides infection, numerous situations lead to the presence of aberrant host DNA in the cytoplasm leading to the activation of the cGAS–STING pathway, for instance in the context of mitochondrial stress, senescence and inflammation [26]. Chromosomal instability, a hallmark of cancer, results from errors during mitotic chromosome segregation and can lead to micronuclei formation, whose rupture leads to cytosolic self-DNA release and cGAS activation [27,28]. The link between DNA damage and STING activation in cancers has been recently reviewed [29]. STING-mediated cytosolic DNA sensing is important for innate immune recognition of immunogenic tumors, for instance by enhancing DCs capability to cross-present antigen [30,31].

Owing to their negative charges, cGAMP and other CDN are unable to passively diffuse across the plasma membrane. cGAMP is transferred from one cell to another through gap junction [32,33], as well as multiple transporters with cell type and species specificities, including the volume-regulated anion channels (VRAC or LRRC8) [34,35], SLC19A1 [32,33] and SLC46A2 [36]. Interestingly, in infected cells cGAMP is packaged within viral particles or extracellular vesicles and is efficiently delivered to target cells promoting innate immunity and antiviral defenses [37]. cGAMP is also transported by the purinergic receptor P2X7 upon apoptotic tumor cell clearance (efferocytosis) blockade promoting STING/type I IFN-mediated antitumor activities [38].

While STING-dependent type I IFN production elicits potent antiviral activities, STING also triggers a variety of cellular processes. In contrast to STING-deficient mice, mice carrying a mutation in STING that impedes type I IFNs are still able to mount protective immune responses against herpes simplex virus 1 (HSV-1) infection [39,40]. An early study showed that *Mycobacterium tuberculosis* DNA triggers STING-dependent autophagy and resistance to infection [41]. It was later confirmed that activated STING binds to the autophagy-inducing protein LC3 in a TBK1- and type I IFN-independent manner, promoting both autophagy and STING degradation to regulate STING-mediated immune activation [42]. Interestingly, STING-dependent autophagy regulation may have emerged earlier than type I IFN induction. Indeed, it has been shown that STING from the sea anemone Nematostella vectensis induces effective autophagy but does not contain the CTT domain that is essential for IRF3 activation and type IFN production [1,7]. STING signaling in T lymphocytes predisposes them to apoptosis [43] and in myeloid cells STING initiates a cell death program upstream of NLRP3 [44]. In T lymphocytes, the LRRC8 protein transports cGAMP resulting in STING activation and p53-mediated apoptosis [45] LRRC8 deficiency enhances T cell-mediated antiviral immunity to influenza as well as to central nervous system inflammation in the experimental autoimmune encephalomyelitis (EAE) model [45]. STING availability is controlled by homeostatic regulation. For instance, STING degradation at steady state is prevented by the stabilizer protein TOLLIP through direct interaction [46].

Of note, there are other receptors for bacterial and host CDNs. For instance, the helicase DDX41 recognizes bacterial CDNs such as cyclic di-GMP and cyclic di-AMP to activate type I IFN-dependent immune response [47]. CDNs also bind to the oxidoreductase RECON, antagonizing STING and NF-κB activation by depleting CDNs availability [48]. Furthermore, c-diAMP and c-diGMP have been shown to induce robust NLRP3 inflammasome activation and IL-1β secretion independently of STING [49].

## 2. STING Agonists

STING agonists include natural bacterial- and host-derived CDNs, chemically modified CDN as well as non-CDN molecules (Figure 1 and Table 1).

### 2.1. Cyclic Dinucleotides-Based STING Agonists

The unique host-derived STING activator is the non-canonical CDN cGAMP synthetized by the enzyme cGAS [51]. cGAMP binding to STING, with high affinity, stimulates the innate immune response in mammalian cells [14,15,50]. An early study showed that cGAMP is an adjuvant that boosts antigen-specific T cell activation and antibody production in mice in the context of a protein immunization model [69].

Bacterial CDNs display critical functions as second messengers in a variety of cellular processes as well as signal transduction, biofilm formation and virulence [70]. Upon bacterial infection, the two most common bacterial CDN activating STING are c-di-GMP and bis-(3′,5′)-cyclic diadenosine monophosphate (c-di-AMP) [71]. C-di-GMP is a universal second messenger in bacteria, regulated by diguanylate cyclases (DGC) and phosphodiesterases (PDE) [63]. C-di-AMP is also a common second messenger molecule in bacteria and archaea, for instance to maintain viability on rich medium [72]. Of note, both c-di-AMP and c-di-GMP have not been found in eukaryotes [73]. It has also been shown that *Vibrio cholerae* produces 3′,5′-3′,5′ cyclic GMP-AMP (3′3′-cGAMP) [74], which induces STING-dependent IFN-β production [51]. Interestingly, anemone cGAS produces a canonical 3′,3′ linked cGAMP similar to those in bacteria [1]. Bacteria also produce other CDNs that do not bind to STING [75]. In short, STING is a direct innate immune sensor of c-di-GMP [12], c-di-AMP [71] and both 2′,3′ and 3′,3′ linkage isomers of cGAMP [13,50,51,76]. Other canonical (3′-5′-linked) CDN have been synthesized; however, they do not show strong STING-dependent IRF induction [77]. Various analogs of c-di-GMP have been synthesized by modifying, for instance, the nucleobase, the sugar residues or the phosphate linkage as recently reviewed [78].

Numerous studies have shown that c-di-GMP or c-di-AMP treatment elicits strong adjuvant properties by promoting DC maturation and T cell priming capacity as well as enhancing antibody production. For instance, in vivo treatment with c-di-GMP potently reduces *Staphylococcus aureus* infection [79] and c-di-AMP has been shown to promote both antibody production as well as cellular immunity [80]. Intranasal c-di-GMP administration induces TNF production by cDC2 inducing Bcl6^+^ monocyte-derived DC (moDCs) differentiation, which promotes memory Th cells in the lungs [81]. Sublingual treatment with 3′3′-cGAMP induces STING-dependent systemic and mucosal immunity against anthrax toxins [82]. Besides myeloid subsets, CDNs also activate B and T lymphocytes directly. B lymphocytes directly respond to CDN stimulation in a STING-dependent manner promoting their activation [83] and a recent study showed that cGAMP directly alters Th1 and Th9 cells differentiation and effector functions [84].

Nevertheless, CDN characteristics, including molecular weight, negative charges and phosphodiester linkage, do not typically match with small molecule drug candidates. Although the bacterial c-di-GMP and c-di-AMP are being developed as potential vaccine adjuvants [85], cGAMP appears as a much more potent STING ligand [50]. In addition, various pathogens, including *Mycobacterium tuberculosis* and *group B streptococcus,* express phosphodiesterases (PDEs) that cleave both bacterial and host-derived CDNs to dampen STING activation. Thus, the use of PDE inhibitors might enhance STING activation and protection against certain infections [86]. cGAMP is hydrolyzed by ecto-nucleotide pyrophosphatase phosphodiesterase 1 (ENPP1) and nonhydrolyzable analogs have been designed [87,88]. Synthetic cyclic adenosine-inosine monophosphate (cAIMP) analogs activate STING with greater affinity as compared to reference compounds and are less sensitive to enzymatic cleavage in vitro [52]. A recent review summarized derivatives based on native CDN structures [89]. For instance, ADU-S100 (ML RR-S2 CDA) displays antitumor efficacy by increasing dendritic cell activation and tumor antigen-specific CD8^+^ T cells [90]. Other modified CDNs such as CDG^SF^, a phosphorothioate and fluorine containing c-di-GMP [53] and a modified c-di-AMP (ML RR-S2 CDA) [54] elicited strong adjuvant function and antitumor activities.

### 2.2. Non-Nucleotide-Based STING Agonists

Several non-nucleotide-based STING agonists have been reported [89]. 5,6-dimethylxanthenone-4-acetic acid (DMXAA) has been shown to induce STING-dependent *Ifnb1* gene expression [56]. In mice, intratumoral DMXAA administration induced robust tumor regression accompanied by a systemic immune response able to promote metastasis rejection [54]. In a murine non-small cell lung cancer (NSCLC) model, DMXAA induces tumor site-specific vascular disruption and an M1 macrophage phenotype [91]. Unfortunately, DMXAA displayed no benefit when used in combination with chemotherapy in a phase III clinical trial in patients with advanced NSCLC [92] and it was later shown that DMXAA is mouse-selective and does not activate human STING [57]. Another non-CDN molecule, a dimer of amidobenzimidazole (diABZI) strongly activates both human and mouse STING to elicit strong antitumor activity [55].

Other non-nucleotide-based STING agonists have been identified, with specific mouse–human binding capacities, including α-mangostin [58], 10-carboxymethyl-9-acridanone CMA [60], dispiro diketopiperzine compound DSDP [61] and CF501 [59].

## 3. STING Antagonists

There are two types of STING antagonists: compounds binding to its palmitoylation sites near the transmembrane domain (Cys88 or Cys91 residues) and others occupying the CDN binding site acting as competitive antagonists (Figure 1 and Table 1).

### 3.1. STING Antagonists Targeting the Palmitoylation Sites

Ablasser’s lab identified nitrofuran derivatives (C-176 and C-178) as covalent small-molecule inhibitors of STING, irreversibly binding to Cys91 [62]. C-176 markedly reduced STING agonist-mediated type I IFNs and IL-6 production in the serum [62]. DNA exonuclease Trex1^−/−^ mice treated with C-176 displayed a significant reduction in serum levels of type I IFNs and a strong suppression of heart inflammation [62]. C-176 treatment reduced proinflammatory cytokine production in a model of LPS-induced acute lung injury (ALI) [93]. Another compound, 3-acylamino indole derivative H-151 inhibits STING-dependent responses in vitro and in vivo [62]. H-151 is active against both human and mouse STING [62]. Endogenous nitro-fatty acids (NO_2_-FAs) covalently modify STING by nitro-alkylation, inhibiting STING palmitoylation of both Cys88 and Cys91 in human and murine cells [64]. NO_2_-FA treatment inhibits pTBK1 and type I IFN production by fibroblasts derived from patients exhibiting a gain-of-function mutation of STING [64]. The acrylamide BPK-25 binds to Cys91 of STING and inhibits cGAMP-mediated STING activation in primary human T cells [63].

### 3.2. STING Antagonists Targeting the CDN-Binding Site

Synthesizing tetrahydroisoquinoline analogs, Siu et al. identified compound **18** as a STING antagonist targeting the CDN-binding site [65]. The plant-derived cyclopeptide astin C binds competitively to the CDN site [66]. In vivo, astin C treatment enhances HSV-1 infection while on the other hand it decreases autoinflammatory responses observed in the absence of the exonuclease Trex1 [66]. SN-011 binding to the CDN pocket maintains STING in an open inactive conformation and inhibited downstream IFN and inflammatory cytokine productions [67]. In vivo, SN-011 treatment limited systemic inflammation and death observed in Trex1^−/−^ mice [67].

### 3.3. Other STING Antagonists

SP23, a bifunctional chimeric protein targeting STING and recruiting the E3 ubiquitin ligase, promotes STING degradation through the ubiquitin-proteasome pathway [68]. SP23 displays anti-inflammatory effects in a mouse model of cisplatin-induced acute kidney injury [68]. VS-X4, a small molecule heterocycle, has been shown to inhibit STING; however, the mechanism of action remains to be determined [94].

## 4. STING and Lung Diseases

A summary of the roles of STING pathway in lung diseases is provided in Table 2.

### 4.1. Autoimmunity

#### 4.1.1. SAVI

STING-associated vasculopathy with onset in infancy (SAVI), classified as an interferonopathy i.e., a disorder associated with an upregulation of interferon, is an autoinflammatory disease with early onset systemic inflammation, skin lesions, failure to thrive and perivascular inflammation associated with high interferon-stimulated gene (ISG) expression profile [96,97]. In most cases, respiratory symptoms, interstitial lung disease (ILD) and pulmonary fibrosis drive premature death [96,135,136,137]. SAVI has been discovered from genetic analysis of the *Tmem173* gene encoding STING from six children in whom three mutations in the exon 5 have been described (N154S, V155M, V147L) [96]. Other variants in *Tmem173* (F153V, G158A and H72N) have been recently described, eliciting key features of the SAVI disease with type I IFN signature in mononuclear cells, although patients display milder clinical manifestations [138]. It has been shown that STING-N154S disrupts calcium homeostasis in T cells, rendering these cells hyperresponsive to T cell receptor signaling-induced ER stress leading to cell death [139].

Owing conserved STING sequences, mouse models of orthologous *Tmem173* gene carrying point mutation corresponding to recurrent mutations observed in SAVI patients have been developed. Heterozygous N153S and V154M mouse strains display lymphopenia and developed IRF3- and type I IFNs-independent severe combined immunodeficiency disease (SCID) occurring early in thymic development [140,141]. Unexpectedly, mild ISGs up-regulation was observed in STING N153S mouse cells as well as in STING N154S SAVI patient fibroblasts [140]. Key features of SAVI were reversed when N153S mice were crossed with RAG^−/−^ and TCR^−/−^ mice but not cGAS^−/−^, IRF3/7^−/−^ or IFNAR^−/−^ suggesting that the SAVI phenotype does not depend on type I IFNs but rather on a T cell-dependent effect [142]. Surprisingly, a recent publication has suggested a contribution for STING-dependent IFN-γ in mouse SAVI and also showed that STING N153S macrophages displayed enhanced *Cxcl9* expression and activation markers [143].

SAVI is an interferonopathy and since type I IFNs signal via the Janus kinase (JAK)/signal transducer and activator of transcription (STAT) pathway, current therapies have been focusing on the use of JAK inhibitors. It was first shown in the original article that JAK inhibitors reduced STAT1 phosphorylation in lymphocytes [96]. JAK1/2 inhibitors (e.g., ruxolitinibb) have since been used with beneficial effects reported on lung morphology and function [144,145,146] as recently reviewed [147]. However, JAK inhibitors may also lead to adverse effects such as enhanced susceptibility to viral respiratory infection [148]. The potential benefit of direct STING inhibitors in SAVI patients remains to be determined.

#### 4.1.2. COPA

A 2015 whole-exome sequencing study identified a rare inflammatory and autoimmune disease caused by autosomal dominant mutation in the *Coatomer protein subunit alpha* (COPA) gene and that was characterized by ILD, high-titer autoantibodies and inflammatory arthritis [149]. COPA is part of the coatomer protein complex I (COPI) important for the retrograde transport of cargo proteins between the Golgi and the ER as well as transit of vesicles between Golgi cisternae [150]. COPA disease was subsequently related to a high type I IFN profile [151]. Mutations in the WD40 domain of COPA (e.g., E241K) lead to a defect in retrograde transport of C-term dilysine-containing protein and thus impaired retrograde intracellular trafficking [149,152]. Of note, STING does not contain dilysin motif itself and the cargo protein SURF4 has been shown to facilitate the interaction between COPI vesicles and STING [153]. Similar to SAVI, COPA syndrome leads to spontaneous STING activation leading to increased type I and type III IFN production as well as downstream ISGs [152,153]. A role for cGAS in STING-mediated COPA syndrome is still controversial, as some studies have shown cGAS dependency [95,152] not all of them [154]. Using a *Copa^E241K/+^* knock-in mouse strain, it has been shown that mutant mice spontaneously develop ILD and that thymic tolerance defect causes thymic epithelial cells-dependent T cell-mediated autoimmunity [155].

COPA syndrome shares clinical features with SAVI, including ILD development. A few dozen patients have been reported worldwide to date and limited clinical data are available regarding therapeutic strategies. Current research has focused on targeting the downstream STING signaling pathway to decrease constitutive type I IFN production. JAK1/2 inhibitor ruxolitinib has been shown to reduce IFN signaling as well as to partially reduce pulmonary disease [156] and strongly reduce rheumatoid symptoms [157]. Peripheral blood mononuclear cells (PBMCs) from a COPA syndrome patient treated with STING inhibitor H-151 have shown reduced IFN-β and ISGs productions [153]. In contrast, while JAK1/3 inhibitor tofacitinib also decreased ISGs, no effect was observed on IFN-β production [153]. A recent study has confirmed that spontaneous STING activation drives the inflammatory response in COPA syndrome due to impaired STING retrograde trafficking and that treatment with H-151 reduced ISG production but not IFN-β [95].

In contrast to SAVI and COPA syndrome, other interferonopathies do not trigger ILD, suggesting that STING intrinsic functions are required for lung disease. The exact contribution of type I IFNs remains to be investigated. In mouse models, lung pathology relies on T cells [140,141,142,155]. A direct therapeutic targeting of STING through the use of inhibitors may be promising.

### 4.2. Infectious Diseases

#### 4.2.1. Coronaviruses

Severe acute respiratory syndrome coronavirus-2 (SARS-CoV-2) is a highly contagious RNA virus responsible for the ongoing 2019 coronavirus disease (COVID-19) pandemic characterized by lung pathology and extrapulmonary complications [158,159]. STING agonists 2′2′-cGAMP and 2′3′-cGAMP are potent antivirals against SARS-CoV-2 infection in respiratory epithelial cells [113]. Authors have also shown that STING agonist diABZI [55] restricts viral replication in primary human bronchial epithelial cells and in mice [113]. Cell fusion caused by the SARS-CoV-2 spike (S) protein to invade host cells induces nucleus damage and micronuclei formation, which are sensed by the cGAS–STING pathway leading to type I IFNs [116]. In contrast, another study has reported that cGAS–STING activation in SARS-CoV-2 infected cells leads to NF-κB but not IRF3-dependent responses and that pharmacological inhibition of STING reduced NF-κB-driven inflammatory immune response in human epithelial cells [94].

Two back-to-back articles showed that a single diABZI treatment strongly enhances survival of SARS-CoV-2-infected K18-hACE2 mice [112,113]. DiABZI potently and transiently induces innate signaling pathways, especially type I and type III IFNs, likely promoting protective antiviral responses [113]. In contrast, a recent study has pointed out the deleterious effects of STING in vivo. Mitochondrial DNA release activated the cGAS–STING pathway in cells including macrophages and endothelial cells promoting type I IFN production and correlating with disease damage and immunopathology [115]. STING inhibition by H-151 reduced severe lung inflammation induced by SARS-CoV-2 in mice and improved survival [115]. Overall, the above data strongly suggest that STING is involved in both viral control as well as with host immune response in a highly context-dependent manner. The conflicting data may reflect specificities of type I IFNs where in most cases early type I IFN response limits virus spread, while a persistent type I IFN signature is associated with deleterious inflammation and poor clinical outcome.

#### 4.2.2. Influenza

Influenza A and B are RNA viruses and are among the most common human respiratory pathogens. Influenza A virus (IAV) triggers cGAS-independent STING-mediated type I IFN production [117]. A subsequent study has shown that M2 protein of RNA viruses such as the influenza virus triggers mtDNA translocation into the cytosol, involving the protein mitochondrial antiviral signaling (MAVS) and inducing cGAS- and DDX41-mediated STING-dependent antiviral responses [118]. Of note, the hemagglutinin fusion peptide (FP), allowing the fusion between viral and endosomal membranes, directly interacts with STING to inhibit its dimerization [117]. Dysfunctional telomeres or age-related immune senescence trigger mitochondrial stress-dependent cGAS–STING activation and increased susceptibility to IAV infection [119]. However, the exact contribution of the STING pathway in IAV infection remains to be determined in vivo. It has been shown that self-DNA-mediated STING activation induced by acute lung injury protects mice from subsequent IAV infection through enhanced type I IFN production [160]. Of note, type I and III IFN responses are suppressed during the course of infection by influenza to limit immunopathology [161].

#### 4.2.3. Tuberculosis

Infection with *Mycobacterium tuberculosis* (Mtb) causes tuberculosis (TB), a major global health issue causing over a million deaths per year worldwide. It has been shown that STING limits bacterial replication by promoting autophagy [41]. Cytosolic Mtb DNA binds to cGAS, eliciting STING-dependent type I IFN production [120] while cGAS-deficient mice display late susceptibility to Mtb infection [123]. In contrast, STING-deficient mice did not show increased susceptibility as compared to control mice [123]. Nevertheless, c-di-AMP produced by Mtb induces STING-dependent autophagy and IFN responses and a c-di-AMP over-producing Mtb strain displayed decreased virulence and increased host survival, indicating STING-mediated control of pathogenicity [121]. These contrasting results may indicate discrepancies in methodology as well as bacterial strain-dependent specificities for instance in terms of mtDNA leakage and type I IFN induction [162].

It has been proposed that phenolic glycolipids produced by Mtb activate the STING pathway inducing CCL-2 production and the recruitment of permissive monocytes [122]. The exact mechanism of STING-mediated CCL-2 production by alveolar macrophages remains to be identified, but appears to be independent of both type I IFNs and ESX-1 secretion system [122].

#### 4.2.4. Streptococcus Pneumonia

*Streptococcus pneumonia*, a Gram-positive extracellular bacterial pathogen, is a leading cause of bacterial pneumonia. While STING-mediated cytosolic DNA sensing is activated during pulmonary S. *pneumoniae* infection, this pathway is dispensable for the initial immune response as well as control of bacterial burden [124]. However, a recent study showed that STING synergizes with the MyD88 pathway to induce late IFN-γ production in the lung in a type I IFN-independent manner, which might play a detrimental function in sustaining inflammation enhancing tissue damage and mortality [125]. We therefore speculate that treatment with STING antagonist at a specific time point of the disease’s course might be beneficial for patients.

#### 4.2.5. Non-Typeable Haemophilus Influenzae (NTHI)

NTHI is a Gram-negative bacterium commonly affecting children and one of the leading causes of acute exacerbations of chronic obstructive pulmonary disease (COPD). It has been recently shown that NTHI DNA induces cGAS–STING-dependent type I IFN induction [126,163]. Follow-up studies are required to decipher the role of STING upon NTHI infection.

#### 4.2.6. Legionella Pneumophila

*Legionella pneumophila* is a Gram-negative facultative intracellular bacterium responsible for the severe pneumonia known as Legionnaires’ disease after inhalation of aerosolized contaminated water droplets and infection of alveolar macrophages. L. *pneumophila*-infected macrophages produce IFN-β in a STING- and IRF3- dependent manner, contributing to ISG induction and bacterial clearance [164]. It has been further confirmed that infected macrophages activate the cGAS–STING pathway leading to type I IFN production [120]. In vivo, L. *pneumophila*-infected cGAS- and STING-deficient mice display increased bacterial loads as compared to wild-type controls [165]. Interestingly, the authors showed that the HAQ STING variants, which include three common non-synonymous single nucleotide polymorphisms (R71H-G230A-R293Q), are associated with Legionnaires’ disease [165]. Of note, DNA from bacteria including L. *pneumophila* as well as *Francisella tularensis* and *Listeria monocytogenes* are excreted into extracellular vesicles from infected cells and delivered to bystander cells to amplify cGAS–STING-dependent pathway [166].

### 4.3. Inflammatory and Allergic Diseases

#### 4.3.1. Allergic Diseases

Asthma is a common respiratory condition and a major public health concern, characterized by chronic lung inflammation, mucus hypersecretion, airway remodeling, hyper-responsiveness and reversible airway obstruction. Asthma is triggered by a variety of allergens including house dust mite (HDM) and air pollution. Mouse models of asthma are generally performed by exposing mice to an allergen together with an adjuvant, historically aluminum salts (alum), inducing a well-established T helper 2 (Th2) immune response. Alum induces cell death and host DNA release, which acts as a potent endogenous immunostimulatory signal mediating adjuvant activity [167]. DNase I treatment to digest extracellular DNA decreases antigen-specific CD4^+^ T cells and humoral responses in OVA and alum-treated mice [98,167]. Antigen-specific CD4^+^ T cell priming and IgE are both reduced in STING-deficient mice immunized with OVA/alum as compared to immunized WT mice [98]. Interestingly, these defects are independent of type I IFN signaling as type I IFN-deficient mice do not show decreased antigen-specific CD4^+^ T cell priming or IgE production, suggesting alternative functions of STING in promoting Th2 responses [98]. In contrast, the chitin-derived adjuvant chitosan induces mitochondrial stress and mtDNA release, activating the cGAS–STING–type I IFN pathway to promote Th1 responses [168,169]. Thus, various adjuvants selectively promote polarized immune responses partly through self-DNA release acting as a danger signal triggering cGAS and STING activation.

In parallel, another set of studies incorporated STING agonists themselves as adjuvants. For instance, cGAMP, together with house dust mite (HDM) antigen, enhances IL-33-dependent asthma, characterized by higher levels of HDM-specific serum IgG1 and total IgE as well as increased eosinophils in the airways [100]. In contrast, cGAMP strongly attenuates Th2-associated lung immunopathology and airway hyperreactivity induced by IL-33 or Aspergillus flavus used as a fungal allergen [101]. Mechanistically, cGAMP activates STING-dependent type I IFN production in alveolar macrophage inhibiting IL-33-mediated activation of type 2 innate lymphoid cells (ILC2) in vivo [101]. cGAMP also directly suppresses ILC2 proliferation and function in both human and mouse ILC2 cells in vitro [101]. It was further shown that intranasal c-di-GMP administration suppresses ILC2s and type 2 lung inflammation, while promoting ILC1s expansion and activation following either Alternaria or IL-33 exposure in a STING–type I IFN-dependent manner [102]. In contrast, cGAMP treatment in HDM-sensitized WT mice increased total HDM-specific IgE levels by promoting T follicular helper cell (Tfh) responses [99].

Chronic rhinosinusitis with nasal polyps (CRSwNP) is a common disease inducing type 2 inflammatory responses that shares similar pathophysiology with asthma. It was shown that STING expression is reduced within eosinophilic nasal polyps leading to decreased type I IFN and suppressor of cytokine signaling 1 (SOCS1) expressions leading to increased IL-13 signaling in epithelial cells and thus exaggerated eosinophilic inflammation [103].

Together, STING-mediated responses are highly context-dependent and their targeting may offer novel therapeutic opportunities to alleviate allergic diseases such as asthma.

#### 4.3.2. Chronic Obstructive Pulmonary Disease (COPD)

COPD, the third leading cause of death worldwide, is a disease usually characterized by progressive airway limitation typically driven by various degrees of chronic obstructive bronchitis and emphysema [170,171,172]. COPD is predominantly caused by cigarette smoking [173]. Using an acute model of cigarette smoke exposure in mice, we showed that it induces self-DNA release in the alveolar space activating a cGAS–STING-dependent neutrophilic influx and inflammatory response [104]. It has been further shown that targeting extracellular self-DNA using a DNAse I treatment alleviates cigarette smoke-induced lung inflammation, notably neutrophil extracellular traps (NETs) and neutrophil-associated proteases [105]. Of note, aerosolized recombinant human DNase I is currently used in patients with cystic fibrosis (CF) [174,175]. In contrast, sub-chronic cigarette smoke exposure lowers STING lung expression limiting subsequent immune response to infection [106]. STING expression in bronchial and lung tissues of COPD patients is unaltered as compared to control groups [176].

COPD is punctuated by life-threatening clinical exacerbations mainly elicited by bacterial (e.g., *Haemophilus influenzae*, *Streptococcus pneumoniae* and *Pseudomonas aeruginosa*) and viral (e.g., Rhinovirus, Coronavirus and Influenza virus) infections [171,172,177]. It has been shown recently that inactivated P. *aeruginosa* PAO1 vaccine stimulates the cGAS–STING pathway and protects elastase-induced COPD mice against subsequent PAO1 infection [107]. In addition, the cGAS–STING signaling pathway is also activated upon *Nontypeable Haemophilus influenza* (NTHI) infection upon bacterial DNA release [126].

#### 4.3.3. Fibrosis

Idiopathic pulmonary fibrosis (IPF) is the most common and severe type of interstitial lung disease (ILD) characterized by dysregulated alveolar repair leading to pathological lung scarring [178,179]. IPF physiopathology relies on repeated lung micro-injuries leading to DNA damage and cell death triggering dysregulated tissue repair and fibrosis. Employing the classical murine model of human IPF by airway exposure to bleomycin (BLM), we showed that STING plays a protective role in limiting fibrosis in an unexpected type I IFN-independent manner [108]. Our data are in line with a study in IPF patients showing that STING expression in blood immune cells correlates with clinical improvement during acute exacerbation [180]. Of note, elevated mtDNA copy numbers in the plasma of IPF patients was used as a biomarker to predict death [181]. Silicosis, another type of pulmonary fibrosis, is caused by chronic inhalation of silica particles associated with increased risk of cancer. In mice, airway silica exposure triggers self-DNA release leading to STING-dependent type I IFN responses promoting lung inflammation [109].

Nanomaterials such as carbon nanotubes (CNTs) and their related compounds display remarkable properties with a wide range of medical and non-medical applications. However, their safety raises concern and occupational disease such as pulmonary fibrosis. It has been shown that airway exposure to graphitized multi-walled carbon nanotubes (GMWCNTs) activates the cGAS–STING pathway and that treatment with the STING inhibitor C-176 decreases pulmonary inflammation and fibrosis in mice [111]. Airways treatment with low dose STING agonist diABZI has been shown to induce cell death by PANoptosis and DNA-mediated acute respiratory distress syndrome (ARDS) [110], a risk factor for progression to fibrosis. Together, the exact contribution of STING in lung fibrotic processes remains to be determined and is likely highly context dependent.

### 4.4. Cancer

Lung cancer, either small cell lung carcinoma (SCLC) or non-small cell lung carcinoma (NSCLC), is one of the most common types of cancer malignancies and remains an important public concern with low five-year survival rate and accounting for about 25% of all cancer deaths in the United States [182]. In addition to primary cancer, the lungs are also one of the most common sites of tumor metastases [183]. The role of the STING pathway in the cancer field is extensively studied with massive implication in immunotherapy and numerous clinical trials have been performed [184]. Tumor-derived DNA activates the cGAS–STING pathway in tumor-infiltrating DC inducing type I IFN production upregulating CD8^+^-mediated antitumor activity [31]. cGAS–STING expression levels are elevated in lung adenocarcinoma in a stage-dependent manner and higher expression correlates with localized adenocarcinoma and overall survival [130]. In contrast, another report showed that nuclear cGAS suppresses DNA repair and promotes tumorigenesis [129]. These conflicting results are likely to reflect the heterogeneity of the anti-tumor responses, for instance in terms of microenvironment and immune status.

Immunotherapies usually target the immune checkpoint blockade (ICB) through the use of monoclonal antibodies against inhibitory signaling molecules expressed on tumor and immune cells, such as programmed death-1 (PD-1), PD-1 ligand (PD-L1), and cytotoxic T-lymphocyte associated protein 4 (CTLA4). NSCLC patients with a high STING pathway activation pattern display higher levels of targetable immune checkpoints and markers of active immune microenvironment associated with positive immunotherapy responses [131]. However, only a fraction of SCLC patients respond to ICB and research is ongoing to develop new drug combinations to enhance its antitumor efficacy, for instance in targeting the DNA damage response (DDR) pathway. DDR inhibition by pharmacological inhibitors of poly ADP-ribose polymerase (PARP) or checkpoint kinase 1 (CHK1) leads to cytosolic DNA activating the cGAS–STING pathway in SCLC cell lines and tumors [134]. In vivo treatment with DDR inhibitors in combination with anti-PD-L1 leads to a strong anti-tumor effect in a cGAS–STING-dependent manner [134]. In addition, DNA repair protein excision repair cross-complementing group 1 (ERCC1) is frequently impaired in NSCLC. It has been shown that PARP inhibition using clinically approved drugs leads to micronuclei formation activating the cGAS–STING pathway in ERCC1-defective NSCLC [132].

*KRAS* belongs to the canonical RAS family of genes and mutation in *KRAS* is a common oncogenic event in lung cancer. *KRAS*-driven cancers frequently inactivate liver kinase B1 (LKB1) and remain largely refractory to most available treatments [185]. It as been recently shown that LKB1 loss is associated with decreased STING expression in KRAS mutant lung cancer resulting in impaired T cell recruitment and antitumor activities. Overexpressing STING restores PD-L1 T cell chemotaxis. Thus, strategies to restore STING expression may have significant therapeutic benefit [133]. In contrast, another study showed that STING-triggered indoleamine 2,3 dioxygenase (IDO) activity in the tumor microenvironment (TME) promotes Lewis lung carcinoma (LLC) growth and that STING deficiency led to increased CD8^+^ T cell-mediated tumor cell killing indicating that STING decreases CD8^+^ T cell effector functions in this context [128]. Together, these results and others show that promoting the STING pathway may have an opposite outcome during tumorigenesis, reflecting context- and kinetic-specific discrepancies.

Radiation-induced antitumor effects have been linked to self-DNA leakage, cGAS–STING signaling and protective IFN-β [30]. However, caution is necessary as, in a mouse model, high radiation doses have been shown to induce DNA exonuclease TREX1 decreasing cGAS activity, while, in contrast, repeated irradiation at lower doses does not induce TREX1 expression allowing robust cGAS–STING activation [186].

DNA damage induced by a low dose of the chemotherapy agent carboplatin activates the STING signaling pathway and synergizes with PD-1 inhibitors to promote protective CD8^+^ T cell infiltration in lung cancer [127].

Treatment with a synthetic c-di-AMP derivative has shown tremendous efficacy against lung metastases, revealing its translational potential as a cancer therapeutic [54]. Intravenously injected c-di-GMP-loaded lipid nanoparticles overcome anti-PD-1 resistance in melanoma lung metastasis via NK cell-mediated IFN-γ production, together with increased PD-L1 expression in cancer cells [187]. Therapeutic vaccines incorporating STING agonist cGAMP, TLR9 ligand CpG oligonucleotide and tumor antigen peptides within nanoporous microparticles are effective in inhibiting lung metastatic melanoma and primary breast cancer by inducing type I IFN in DCs, promoting CD8^+^ and CD103^+^ DC maturation and priming capacity [188]. Aerosol inhalation of chitosan/anti-PD-1 nanocomplex has been shown to activate the cGAS–STING pathway in DCs, leading to type I IFN-dependent DC activation and lung metastasis regression [189].

## 5. STING in Lung Vaccine/Adjuvant Formulation

Pathogen-associated molecular pattern (PAMP) recognition by pathogen recognition receptors (PRRs) provides the molecular basis of innate immune activation in adjuvant functions. Exogenous adjuvants are typically critical in various vaccines, including subunit vaccines, however natural adjuvants such as host-derived “danger signals” also elicit adjuvant properties. Vaccines incorporating STING agonists as an adjuvant elicit a robust immune defense against infections and cancer (Table 3). A single dose of the natural STING agonist cGAMP as a mucosal adjuvant effectively protects mice against a lethal dose of influenza virus by boosting CD4^+^ and CD8^+^ T cells responses [190]. Mice vaccinated intranasally with an influenza hemagglutinin (HA) vaccine with cGAMP display increased germinal center formation and IgA production in nasal-associated lymphoid tissue [191].

Other STING agonists, such as the bacterial c-di-GMP also display strong mucosal adjuvant activity. Of note, c-di-GMP’s properties as a potent adjuvant were described before STING discovery [79,202,203]. It was later shown that c-di-GMP-adjuvanted vaccine generates better protection against *Streptococcus pneumoniae* infection as compared to cGAMP by inducing pinocytosis in DCs, stronger T helper cell polarization and effector responses and higher antibody responses [199]. Interestingly, type I IFN signaling is not required for the mucosal adjuvant activity of c-di-GMP in vivo but requires NF-κB-dependent TNF-α to induce antigen-specific antibody and Th1/Th2 cytokine production [204]. Of note, c-di-GMP does not induce IFN-β but induces IFN-λ [199]. Nevertheless, cGAMP-adjuvanted inactivated H7N9 vaccine conferred strong protection against homologous infection and cross-protection against H1N1, H3N2, and H9N2 influenza viruses by stimulating both humoral and cellular immune response [190]. C-di-AMP-based vaccination has been shown to trigger STING-dependent type I IFN production inducing cytotoxic T lymphocyte responses and protective immunity [192].

However, STING agonist delivery to the cytoplasm remains a limitation and its encapsulation may be important. For instance, it has been shown that 3′3′-cGAMP encapsulation in dextran-based polymeric microparticles elicited stronger humoral and cellular immune responses [193]. Single cutaneous vaccination with influenza hemagglutinin (HA) and cGAMP enhances survival upon IAV challenge by promoting cellular and humoral immune responses [194]. A lower dose of cGAMP is required when encapsulated in microparticles and given as an electrohydrodynamic spraying formulation, providing long-term protection against lethal influenza challenge [193]. cGAMP-containing liposomes also induce long-lasting cross-protection against subsequent heterosubtypic influenza infection by triggering STING activation in alveolar epithelial cells inducing humoral and CD8^+^ T cell immune responses [195]. Intranasal vaccination with encapsulated cGAMP within a subunit vaccine composed of negatively charged liposomes and adsorbed spike protein elicited systemic and mucosal immunity against SARS-CoV-2 by providing both T cell responses and neutralizing antibodies [198]. A combination of a new non-nucleotide small molecule STING agonist CF501 and a subunit vaccine consisting of SARS-CoV-2 receptor-binding domain (RBD) of the spike protein of SARS-CoV-2 and Fc fragment of human IgG induces a strong SARS-CoV-2 neutralizing antibody response [59]. In addition, SARS-CoV-2 spike protein immunization with a synthetic CDN (CDG^SF^) induces high antibody titer and a robust T cell response [53]. In the context of Mtb, mucosal administration of protein subunit vaccine including various common antigens with synthetic analog of cyclic diguanylate elicits a protective Th1/Th17 immune response in mouse model [196,197]. The protection requires STING but not type I IFN signaling [196]. Another synthetic CDN (ML-RR-cGAMP) appears to display unique properties as compared to other Th17 promoting mucosal vaccines highlighting the strong potential of CDNs as adjuvants for tuberculosis vaccines [197].

In cancer settings, c-di-GMP-loaded liposomes show increased antitumor activity mediated by NK cells in a lung metastatic mouse model with B16-F10 melanoma [187]. Uptake of inhaled phosphatidylserine-coated liposome incorporating cGAMP by APC activates STING pathway and CD8^+^ T cell cross-priming, which synergizes with radiotherapy to elicit anti-tumor activity in B16-OVA melanoma lung metastasis model [201]. A synthetic CDN (ML RR-S2 CDA), which activates both mSTING and hSTING, induces lasting immune-mediated antitumor activity in various models including B16-F10 melanoma cell-triggered lung metastases [54]. In a murine NSCLC model, DMXAA induces vascular disruption in the tumor environment and a M2 to M1 macrophage repolarization [91]. Systemic administration of cGAMP encapsulated within polyethylene glycol-coated cationic liposomes leads to a strong increase of *Ifnb1* and *Cxcl9* expressions in the lungs and partially reduces lung metastatic foci size [200].

## 6. Conclusions

Over the last years, tremendous progress has been made in our understanding of STING biology. It is now clear that STING leads to the activation of multiple pathways with cell and tissue specific outcomes. Intense research is ongoing to develop agonists and antagonists with diverse clinical applications, likely reflecting their strong potential as immunomodulators. Antagonists may selectively limit inflammation and display beneficial effects in the context of autoimmunity and lung inflammatory diseases. On the other hand, vaccine incorporating STING agonists as adjuvants showed promising results for cancers and infectious diseases. Numerous STING agonists, e.g., MK1454, E7766, have not been covered here and are already in clinical trials in cancer therapy [205]. Some of the strong limitations of first-generation agonists, including their poor pharmacokinetic and physiochemical properties, might now be at least partially overcome with new synthetic compounds and ad-hoc formulation. However, potential side effects especially in case of systemic STING agonist delivery remain of concern.

Whereas type I IFN is undoubtedly a major downstream effector of STING signaling, various studies demonstrated type I IFN-independent role for the STING pathway, however the mechanisms involved remain elusive. The role of STING in promoting autophagy and cell death regulation is of particular interest as it might be linked to tissue repair and inflammation resolution notably in chronic lung diseases. However, in several lung disease settings, targeting the STING pathway led to opposite results as illustrated recently in mouse models of SARS-CoV-2 infection. Thus, further studies are necessary to characterize the exact mechanisms downstream of STING activation and delineate its potential use as a therapeutic target.

## Figures and Tables

**Figure 1 cells-11-03483-f001:**
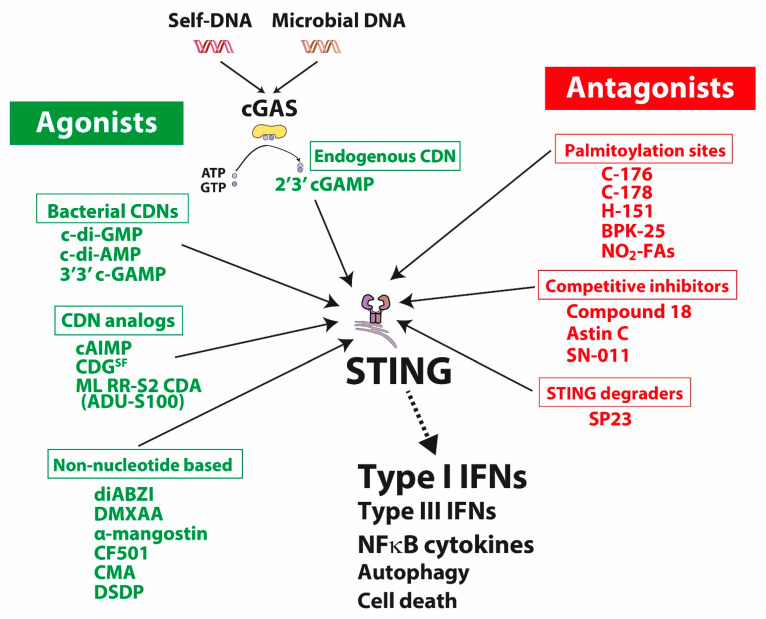
Misplaced self-DNA or microbial DNA in the cytoplasm binds to cGAS producing the cyclic dinucleotide (CDN) 2′,3′-cyclic GMP-AMP (2′,3′-cGAMP) from ATP and GTP. cGAMP binds to STING located in the endoplasmic reticulum (ER) membrane, which undergoes a conformational shift inducing a phosphorylation cascade leading to the activation of genes encoding for type I/III IFNs as well as NF-κB-dependent genes. Activated STING is also directly involved in cellular processes notably autophagy and cell death regulations. Besides 2′,3′-cGAMP as the endogenous STING activator, other agonists include bacterial CDN as well as synthetic compounds, CDN analogs or non-nucleotide-based molecules. STING inhibitors comprise compounds that target its palmitoylation site, act as competitive inhibitors or promote its degradation.

**Table 1 cells-11-03483-t001:** Species specificities of the STING agonists/antagonists.

Agonists		Mouse	Ref.	Human	Ref.
Endogenous CDN	2′3′ cGAMP	YES	[50,51]	YES	[50,51]
Bacterial-CDNs	c-di-GMP	YES	[12]	YES	[12]
	c-di-AMP	YES	[12]	YES	[12]
	3′3′ cGAMP	YES	[50]	YES	[50]
CDN analogs	cAIMP	YES	[52]	YES	[52]
	CDG^SF^	YES	[53]		
	ML RR-S2 CDA (ADU-S100)	YES	[54]	YES	[54]
Non-nucleotide based	diABZI	YES	[55]	YES	[55]
	DMXAA	YES	[56]	NO	[57]
	α-mangostin	(YES)	[58]	YES	[58]
	CF501	YES	[59]		
	CMA	YES	[60]	NO	[60]
	DSDP	NO	[61]	YES	[61]
Antagonists		Mouse	Ref.	Human	Ref.
Palmitoylation sites	C-176	YES	[62]	NO	[62]
	C-178	YES	[62]	NO	[62]
	H-151	YES	[62]	YES	[62]
	BPK-25			YES	[63]
	NO_2_-FAs	YES	[64]	YES	[64]
Competitive inhibitors	Compound **18**			YES	[65]
	Astin C	YES	[66]	YES	[66]
	SN-011	YES	[67]	YES	[67]
STING degrader	SP23	YES	[68]	YES	[68]

**Table 2 cells-11-03483-t002:** Role of the STING pathway in lung diseases.

Category	Disease/Model	Species	Trigger/Pathway			Main Effects/Findings	Ref.
				STING Agonist	STING Antagonist		
Autoimmunity	COPA	H	Coatomer protein complex dysfunction			STING-dependent inflammation, with varying degree of interstitial lung disease	[95]
	COPA	H (Φ), M (Φ)	Coatomer protein complex dysfunction		H-151	H-151 reduces IFN-β and ISG inductions	[95]
	SAVI	H	Gain of function mutation in STING			Interferonopathy associated with skin lesions, perivascular inflammation and interstitial lung disease	[96,97]
Inflammatory diseases	Asthma (OVA/ALUM)	M	Self DNA release			STING deficiency leads to IFNAR independent reduction of antigen specific CD4+ T cell priming and IgE	[98]
	Asthma (HDM)	M		cGAMP		cGAMP increases HDM-specific IgE levels by promoting T follicular helper cells (Tfh) responses	[99]
	Asthma (HDM)	M		cGAMP		cGAMP increases IL-33-dependent asthma and Th2 responses	[100]
	Asthma (IL-33 or Aspergillus flavus)	M		cGAMP		cGAMP decreases Th2-associated lung immunopathology and airway hyperreactivity by inhibiting ILC2 cell activation	[101]
	Asthma (Alternaria alternata or IL-33)	M		ci-di-GMP		ci-di-GMP suppresses ILC2s and type 2 lung inflammation, while promoting ILC1s in a STING/type I IFN-dependent manner	[102]
	Asthma (ILC2)	H (Φ), M (Φ)		cGAMP		cGAMP suppresses proliferation and cytokine production of ILC2	[101]
	CRSwNP	H				Reduced STING/type I IFN expressions within eosinophilic nasal polyps leading to IL-13 signaling and eosinophilic inflammation	[103]
	COPD	M	Self DNA release			cGAS/STING-dependent neutrophilic influx and inflammatory response	[104]
	COPD	M	Self DNA release			DNAse I treatment alleviates cigarette smoke-induced lung inflammation	[105]
	COPD	M				Decreased STING lung expression limiting subsequent immune response to infection	[106]
	COPD exacerbation	M	PAO1 vaccine			cGAS/STING-dependent protection to *P. aeruginosa* infection	[107]
	IPF	M	Self-DNA release			STING decreases lung fibrosis in a type I IFN-independent manner	[108]
	Silicosis	M	Self-DNA release			STING-dependent type I IFN responses promoting lung inflammation	[109]
	ARDS	M		diABZI		diABZI induces PANoptosis and promotes ARDS	[110]
	GMWCNTs	M			C-176	C-176 decreases pulmonary inflammation and fibrosis	[111]
Infectious diseases	SARS-CoV-2	M		diABZI		Strong protection from SARS-CoV-2-triggered lethality	[112,113]
	SARS-CoV-2	H (Φ)		cGAMP; diABZI		Inhibition of SARS-CoV-2 replication	[113,114]
	SARS-CoV-2	M			H-151	H-151 reduces severe lung inflammation and improves survival	[115]
	SARS-CoV-2	H (Φ)	Mitochondrial DNA release		H-151	H-151 reduces type I IFN/ISG production and cell death	[115]
	SARS-CoV-2	H (Φ)	Cell fusion-induced micronuclei			Activation of the cGAS/STING/type I IFN pathway	[116]
	SARS-CoV-2	H (Φ)			H-151; VS-X4	H-151 or VS-X4 limit cGAS/STING-driven NF-κB activation and inflammatory immune response	[94]
	Influenza	M (Φ)				cGAS-independent STING-mediated type I IFN production	[117]
	Influenza	H (Φ), M (Φ)	M2 protein-mediated mtDNA release			cGAS- and DDX41-mediated STING-dependent antiviral responses	[118]
	Influenza	M	Aging/senescence-induced mitochondrial stress			cGAS/STING activation and increased susceptibility to IAV infection	[119]
	Influenza	M			H-151	H-151 decreases viral loads and histopathology	[119]
	M. *tuberculosis*	M	M. *tuberculosis* DNA			STING-mediated autophagy decreases bacterial replication	[41]
	M. *tuberculosis*	M	M. *tuberculosis* DNA			cGAS/STING activation leading to type I IFN production	[120]
	M. *tuberculosis*	M	M. *tuberculosis* c-di-AMP			STING-dependent autophagy and type I IFN production limiting virulence and pathogenicity	[121]
	M. *tuberculosis*	Z	M. *tuberculosis* phenolic glycolipids			STING-mediated CCL-2 production and growth-permissive monocyte recruitment in a type I IFN independent manner	[122]
	M. *tuberculosis*	M				In contrast to cGAS deficiency, STING deficiency shows no effect on mouse survival	[123]
	*S. pneumonia*	M				STING is dispensable for initial control of bacterial burden	[124]
	*S. pneumonia*	M				cGAS/STING and MyD88 pathway-mediated late IFN-γ production	[125]
	NTHI	M	NTHI DNA			cGAS/STING-dependent type I IFN induction	[126]
	*L. pneumophila*	M (Φ)				cGAS/STING-mediated bacterial clearance	[120]
Lung cancer	LLC	M	DNA damage by carboplatin			Synergizes with PD-1 inhibitors to promote protective CD8+ T cells infiltration	[127]
	LLC	M	STING-triggered IDO			Promotes tumor growth and limits CD8+ T cell-mediated tumor cell killing	[128]
	LLC	M	Nuclear cGAS suppresses DNA repair			cGAS promotes tumor expansion	[129]
	NSCLC	H				Increased cGAS-STING expression levels correlating with overall survival	[130]
	NSCLC	H				Increased cGAS expression correlates with tumor stage	[129]
	NSCLC	H				STING pathway activation correlates with efficient immunotherapy	[131]
	NSCLC	H, H (Φ)	PARP inhibition-triggered micronuclei			cGAS/STING activation in ERCC1-defective NSCLC	[132]
	NSCLC	H, H (Φ)	STING suppression in KRAS-LKB1 mutant			Decreased STING-mediated tumor cell cytotoxicity	[133]
	SCLC	M	DDR inhibition + anti-PD-L1			cGAS/STING-dependent anti-tumor effect	[134]

ARDS: Acute respiratory distress syndrome; COPA: Coatomer protein subunit alpha; COPD: Chronic obstructive pulmonary disease; CRSwNP: Chronic rhinosinusitis with nasal polyps; DDR: DNA damage response; GMWCNTs: Graphitized multi-walled carbon nanotubes; H: Human; IDO: Indoleamine 2,3 dioxygenase; IPF: Idiopathic pulmonary fibrosis; LLC: Lewis lung carcinoma; M: Mouse; NSCLC: Non-small cell lung cancer; NTHI: Non-typeable haemophilus influenzae; Φ: in vitro data; PAO1: *Pseudomonas aeruginosa* O1; PARP: Poly (ADP-ribose) polymerase; SAVI: STING-associated vasculopathy with onset in infancy; SCLC: Small cell lung cancer; Z: Zebrafish.

**Table 3 cells-11-03483-t003:** Lung vaccines incorporating STING agonists.

Category	Disease/Model	STING Agonist	Antigen/Costimulant	Carrier	Route	Main Effect	Ref.
Infectious diseases	Influenza	cGAMP	HA (H7N9) or inactivated H7N9	PBS	IN	Protection against a lethal dose of influenza virus	[190]
	Influenza	cGAMP	HA	PBS	IN	Increased germinal center formation and IgA production	[191]
	Influenza	c-di-AMP	OVA-expressing H1N1	PBS	IN	Increased CTL immune memory and reduced weight loss upon viral challenge	[192]
	Influenza	3′3′-cGAMP	HA (H1N1)	Acetalated dextran polymeric microparticles	IM	Protective immunity against a lethal influenza challenge	[193]
	Influenza	cGAMP	HA (H1N1)		ID (not IM)	Protective immunity against a lethal influenza challenge	[194]
	Influenza	cGAMP	Inactivated H1N1, H5N1, H7N9	Pulmonary surfactant biomimetic liposomes	IN	Protective immunity against a lethal influenza challenge	[195]
	M. *tuberculosis*	RR-CDG, ML-RR-cGAMP	5Ag	AddaVax	SC	Type I IFN-independent Th1 immune response and protection	[196]
	M. *tuberculosis*	RR-CDG, ML-RR-cGAMP	5Ag	PBS	IN	Th17 immune response and enhanced protection	[196]
	M. *tuberculosis*	ML-RR-cGAMP	5Ag or H1	PBS	IN	IL-17-dependent Protection	[197]
	SARS-CoV-2	cGAMP	Spike protein	Negatively charged liposomes	IN	Increased B and T cell responses	[198]
	SARS-CoV-2	CF501	RBD-Fc protein	PBS	IM	Long term immunity against SARS-CoV-2 challenge	[59]
	SARS-CoV-2	CDG^SF^	spike protein		SC	Increased IFN-γ and SARS-CoV-2 specific IgG	[53]
	S. *pneumoniae*	c-di-GMP (>cGAMP)	PspA		IN	Enhanced antigen uptake and protection	[199]
Lung Cancer	Lung metastases (melanoma, breast, colon)	DMXAA; ML RR-S2 CDA		NaHCO_3_; HBSS	IT	Systemic antitumor immunity	[54]
	Lung adenocarcinoma, lung metastasis (breast)	DMXAA		DMSO	IP	M1 macrophage polarization-associated antitumor immunity	[91]
	Lung metastases (melanoma)	cGAMP	Anti-CTLA-4 and anti-PD-1 antibodies (IP)	PEG-containing cationic liposomes	IV	Synergistic antitumor immunity	[200]
	Lung metastases (melanoma)	c-di-GMP	Anti-PD-1 antibody (IP)	Lipid nanoparticle	IV	NK cell-dependent synergistic antitumor effect	[187]
	Lung metastases (melanoma, breast)	cGAMP	Radiotherapy	Phosphatidylserine coated liposome	IN	Synergistic antitumor immunity	[201]
	Lung metastases (melanoma)	cGAMP	CpG and tumor antigen peptides	Nanoporous microparticles		Increased DC maturation and enhanced survival	[188]
	Lung metastases (melanoma)		Chitosan and anti-PD-1 antibody (aerosol)		IN	cGAS–STING–Type I IFN pathway enhancing DC activation and metastasis regression	[189]

5Ag: Fusion protein containing five M. tuberculosis proteins; CDG^SF:^: c-di-GMP unilaterally modified with phosphorothioate and fluorine; DMXAA: 5,6-dimethylxanthenone-4-acetic acid; H1: Fusion protein of the antigens Ag85B and ESAT-6; HA: Hemagglutinin; IN: Intranasal; IT: Intratumoral; ML RR-S2 CDA: Dithio-(RP, RP)-[cyclic[A(2′,5′)pA(3′,5′)p]]; PEG: Polyethylene glycol; Pspa: Pneumococcal surface protein A; RBD-Fc protein: Receptor-binding domain in the spike protein + Fc fragment of human IgG; RR-CDG and ML-RR-cGAMP: synthetic CDNs; SC: subcutaneous.

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
