# Peer review of "STING Targeting in Lung Diseases"

_cells, 2022, doi:10.3390/cells11213483_

Round 1

Reviewer 1 Report

I enjoyed reading the review manuscript- it is well written and thorough. Overall, the review provides essential information for signaling of STING, STING agonists/antagonists, roles of STING in lung diseases, delivery approaches in clinic. Due to a large body of work and the broad scope of the review, there are some concerns that would need to be addressed before consideration for publication.

Line 50-51: should be phage instead of viral infection.

Ref 20-22: The authors should provide speculations on why STING exerts opposite function upon bacterial infection. Current format reads confusing.

Line 66-71: DNA damage has also been linked to STING activation- that should be included.

Line 71: there are multiple cGAMP transporters and uniporters reported that authors should include.

Line 72-89: Nan Yan group in UTSW has published a series work elucidating non-canonical function of STING in regulating T cell function including calcium regulations. The authors would need to include these findings.

Line 161: should be agonists.

Given mouse STING and human STING are not well conserved in sequence, and previous studies showed DMAXX activates mouse but not human STING, thus, it will be necessary to further indicate if the antagonists in Fig 1 are for human/mouse STING or both, as well as from line 173-175.

Section 2: There is a recent publication on STING-PROTAC that the authors should include in this section as another type of antagonists.

Table 1: Ref 163: There is an earlier report (PMID: 30356214) showing increased cGAS suppresses HR and promotes lung cancer growth- which is not consistent with ref 163. This would need to be added and discussed.

Section 3 line 31: Given the authors stated that STING exerts its function in SAVI independent of innate immunity but through T cell regulation, the authors would need to clarify how STING inhibition would benefit SAVI treatment without affects innate immune function?

Table 1: it will be beneficial if the authors could further clearly indicate in this table innate immunity dependent and independent diseases (eg by different colors).

Conclusion sections is pretty short and should be extended with more speculations on missing information on roles of STING in lung diseases as well as future directions for both research and therapy.

Reviewer 2 Report

Manuscript by Rodrigues et al. presents the biological role of STING pathway in the context of lung diseases. It is a tremendous, very solid work and an in-depth study, gathering in one place a lot of information about involvement of STING in etiology of lung diseases and presenting potential treatment modalities.

My main reservation is though the accessibility of the manuscript.

Numerous works discussing STING functioning are available, and, true, there is no need to repeat the basic points. However, I’m afraid that if a reader comes from the lung disease field and is not all too familiar with cell biology/cell signaling, some aspects are not clear, and the manuscript is overall hard to follow. The information is there (indeed), but not very transparent. If I were a medical doctor working on my PhD and came across this review, I would be overwhelmed by the amount of information, which I would try to memorize, while some aspects can be easily absorbed by simply understanding the general principles.

I think the review may profit from slightly changing its angle from just presenting the multiple facts to more focus on explaining processes as well as making the structure a little bit more intuitive.

For now, the flow is as follows: STING structure à STING pathway à STING ligands being mostly CDNs à the source of CDNs are bacteria, and (sounds more like mentioned a bit a propos) CDNs are by the way also produced in response to pathogen-derived cytosolic double stranded (ds) DNA.

In my opinion it should be stressed (and mentioned early in the manuscript) that the primary role of STING pathway is sensing cytoplasmic DNA. This is not a physiological condition, and therefore such mechanism even evolved (mostly as response to foreign, eg. viral DNA). We profit from it now (or attempt to) in cancer therapy, since presence of cytoplasmic DNA may result from DNA damage caused by radio- or chemotherapy. Thus, we harness a very primitive defense mechanism, and thus cGAS/STING pathway is a link connecting DNA damage to immune system activation.

A brief explanation of how the immune system is activated (including for instance the mechanism of how STING pathway contributes to antigen presentation, which also partly explains its role in cancer treatment) would also be welcome.

Minor comments

Information about in which types of cells STING pathway operates is missing (with special focus on lungs).

Explanation why cGAMP is regarded as non-canonical should appear the first time the term is used or even when cGAMP itself is described.

There seems to be something off with the CDN description. Why do the Authors even give a description of a cyclic nucleotide? Why is it relevant if the fact that the cyclic dinucleotide even is cyclic comes from the presence of TWO nucleoside monophosphates?

Also, I appreciate that the Authors give some stage light to CDNs being of bacterial origin rather than exclusively synthetized by cGAS. However, it should be explained how STING and CDNs find each other, in case CDNs are produced by extracellular pathogens and STING resides inside the cell. What happens then? Phagocytosis of bacteria by macrophages or DCs and thus STING activation inside immune cells? CDNs penetrating the cell membrane?

Line 127

(NB. Line numbering is not exactly consecutive as it starts over again on page 9 and then page 18 (seemingly after each table). I hope the Authors will be able to identify which is the line in question, as the comments may not always put in a sequential order.)

How DNA from S. pneumoniae (being an extracellular pathogen) becomes “cytosolic DNA” during infection?

Line 177

Could the Authors briefly explain what the “palmitoylation sites” are what their role is?

Line 2 and below in 3.1 Autoimmunity chapter

Could the authors explain that the term “interferonopathy”? It would facilitate understanding the following section and the role of STING mutations in SAVI (at first, I automatically assumed they inactivate STING).

Line 164

How does the “host DNA release” (into the extracellular space?) after cell death become an “endogenous immunostimulatory signal”?

Line 169 and below

“(…) these defects are independent of type I IFN signaling, suggesting alternative functions of STING in promoting Th2 responses”

This is a good example of a detailed fact presented to be memorized, but not accompanied by any explanation. I am not sure what happened in this paragraph. I honestly do not know how Th2 response came into the picture, I am not sure which defects are “these defects” (line above mentions STING-deficient mice, but I struggle to understand how come it is IFN I independent and if “alternative functions of STING” can be found in STING-deficient mice). This fragment is definitely off.

Below there are also statements such as: “In contrast, cGAMP treatment in HDM-sensitized WT mice increased total HDM-specific IgE levels by promoting T follicular helper cells (Tfh) responses (137).” It is undoubtedly true, but after reading this sentence (and basically the whole paragraph) I still don’t know if I want my asthma to be treated with cGAMP or better not. None of the works cited above (most of them rather complex) is accompanied by information if the cGAMP would alleviate the symptoms or make them more severe. Instead of being lead gently through the review to get the main points and avoid laborious digging into the original articles the reader has to struggle real hard to get what the Authors meant.

Line 203

Again, it is not clear how self-DNA released into the alveolar space activates cGAS/STING. I honestly think it is due to neutrophils clearing dead cells, but the article does not provide the information. Same with graphitized multi-walled carbon nanotubes. I’m also guessing it is a secondary effect due to cell damage.  

Line 271

“KRAS belongs to the canonical RAS family of genes and mutation in KRAS is a common oncogenic event in lung cancer. KRAS-driven cancers frequently inactivate liver kinase B1 (LKB1) and remain largely refractory to most available treatments (167). It was recently shown that LKB1 decreases STING expression in KRAS mutant lung cancer leading to impaired T cell recruitment”

This fragment is not logical, and I actually think it comes from misunderstanding of the original work.

1.       KRAS-driven cancers frequently inactivate liver kinase B1 (LKB1)

2.       LKB1 decreases STING

3.       impaired T cell recruitment

That would suggest that LBK1 is “the bad guy”, but it wouldn’t explain, why, if in KRAS driven cancers the LBK1 is decreased, the STING would be inhibited as well

What Kitajima et al. (2018) actually say is: “…we provide the first evidence that STING is actively suppressed following LOSS of the LKB1 tumor suppressor gene.” So, it’s not LBK1, but the loss of LBK1 that inhibits STING.

Line 286 and below

“However, caution is necessary as in a mouse model high radiation doses induces DNA exonuclease TREX1 decreasing cGAS activity, (…)” This fragment is an example of a very simple and logical process, which is though presented in a “to be memorized way”. TREX1 decreases cGAS in an indirect way – by digesting its substrate. Therefore, high radiation à high TREX1 expression (to get rid of DNA fragments) à low amounts of cytosolic DNA à low cGAS/STING activity à low INF production (eg. doi: 10.1038/ncomms15618).

Along the same lines, DDR inhibition (described in lines 263-270) also works by increasing the amount of cytosolic DNA.

Line 47

“Sting” is not capitalized

Abbreviations

Abbreviations used more than once (eg. ILD, ISGs) can perhaps be explained in a separate section for readers’ convenience. OVA is, as far as I can see, not explained at all. On page 18, line 37 CDN probably refers to a synthetic CDN (CDNSF), mentioned just before. Or is it a remark referring to any CDN?

Round 2

Reviewer 2 Report

I would like to thank the Authors of the manuscript entitled “STING targeting in lung diseases“ for their effort put into modifying their work.

In principle I am willing to recommend the manuscript in its current form for publication. My general impression is though that it is still a bit heavy to read.  Mostly it does not work as a stand-alone article, but rather an inspiration for further reading and indication where to find the relevant information. Perhaps it is not possible to do it otherwise with this degree of complexity.

However, in the longer run, reading a text where each sentence is three-line long and packed with specific terminology is not feasible. In general, I think that long and complicated sentences, which require that the reader goes through them multiple times before getting the take-home message, should be avoided. Researchers tend to be busy people and if a review aims at saving the readers’ time by gathering all the information in one place, this information should also be presented in a clear manner. By that I mean especially Part 3.3.1 “Allergic diseases” or lines 349 – 352, regarding c-di-GMP loaded lipid nanoparticles, but there is more.

Moreover, I have few several minor comments which slipped my attention during earlier reading:

Line numbers starting on page 1

Line 43 – sth is off “Owing negative charges cGAMP A CDN consists of (…)”

Line 50 – sth is off (…) cGAMP a concanonical structure (…)” – which has, is characterized by etc.?

Line numbers starting on page 4

Line 65 – “a few dozen patients” – just to clarify: overall? As in the whole world population since the discovery of the mutation?

Line 121 – “translocation in the cytosol” or “into”/”release into the cytosol”? Moreover, MAVS is not explained

Line 124 – perhaps a short explanation what the hemagglutinin fusion peptide is (FP)

Line 166 – looks like literature reference accidently put in the middle of a word (F[141][141][141]ollow-up)

Line 248 – “(…) of COPD patients is? unaltered”. Is it a typo or something the Authors intended to double-check?

Line 353 – I understand that the authors of ref. 188 may casually refer to the ligand as “CpG”. However, is it really the exact term, or is it an “CpG oligonucleotide” as in an oligonucleotide containing CpG site(s)? Otherwise, it sounds as an isolated CpG being injected.

Line numbers starting on page 13

Line 16 – “(..) were described before STING discovery knowing that it activates STING”. That sounds as if the properties were described before the discovery of STING, but already knowing that c-di-GMP activates STING.
